# Lactate-Lactylation Hands between Metabolic Reprogramming and Immunosuppression

**DOI:** 10.3390/ijms231911943

**Published:** 2022-10-08

**Authors:** Lihua Chen, Lixiang Huang, Yu Gu, Wei Cang, Pengming Sun, Yang Xiang

**Affiliations:** 1Department of Obstetrics and Gynecology, Peking Union Medical College Hospital, Chinese Academy of Medical Sciences, Peking Union Medical College, Beijing 100730, China; 2National Clinical Research Center for Obstetric & Gynecologic Diseases, Peking Union Medical College Hospital, Chinese Academy of Medical Sciences, Peking Union Medical College, Beijing 100010, China; 3Fujian Maternity and Child Health Hospital, College of Clinical Medicine for Obstetrics & Gynecology and Pediatrics, Fujian Medical University, Fuzhou 350001, China

**Keywords:** metabolic reprogramming, immune evasion, lactate, lactylation, immunotherapy

## Abstract

Immune evasion and metabolic reprogramming are two fundamental hallmarks of cancer. Interestingly, lactate closely links them together. However, lactate has long been recognized as a metabolic waste product. Lactate and the acidification of the tumor microenvironment (TME) promote key carcinogenesis processes, including angiogenesis, invasion, metastasis, and immune escape. Notably, histone lysine lactylation (Kla) was identified as a novel post-modification (PTM), providing a new perspective on the mechanism by which lactate functions and providing a promising and potential therapy for tumors target. Further studies have confirmed that protein lactylation is essential for lactate to function; it involves important life activities such as glycolysis-related cell functions and macrophage polarization. This review systematically elucidates the role of lactate as an immunosuppressive molecule from the aspects of lactate metabolism and the effects of histone lysine or non-histone lactylation on immune cells; it provides new ideas for further understanding protein lactylation in elucidating lactate regulation of cell metabolism and immune function. We explored the possibility of targeting potential targets in lactate metabolism for cancer treatment. Finally, it is promising to propose a combined strategy inhibiting the glycolytic pathway and immunotherapy.

## 1. Introduction

Cancer immunotherapy is a milestone in the history of the treatment of cancer. The emergence of immune checkpoint inhibitors (ICIs) significantly prolongs the survival time of patients and has become one of the essential treatments for advanced tumors. ICIs, such as anti-programmed cell death-1(PD-1), anti-programmed cell death-ligand 1 (PD-L1), and anti-cytotoxic T lymphocyte-associated antigen-4 (CTLA-4) [1,2,3] made significant progress in amplifying endogenous anti-tumor T cell responses. Nevertheless, it is reported that nearly 85% of cancer patients have innate or acquired resistance to ICIs, which significantly limits their clinical application [4]. A significant barrier is the burdensome metabolic landscape of the tumor microenvironment (TME); it is well known that the metabolic disorder of tumor cells [5] results in a hypoxic, acidic, glucose- and amino acid–deprived environment. An important feature of energy metabolism in tumor cells is known as the “Warburg effect” [6]; it is reported that cancer cells prefer to consume large amounts of glucose through glycolysis rather than favor oxidative phosphorylation, even in a sufficient supply of oxygen; it has been thought that only cancer cells consume large amounts of glucose, but there is growing evidence demonstrating that immune cells, predominantly myeloid cells, also do [7,8]. Interestingly, lactate further promotes the proliferation of tumorigenic immune cells and builds an immunosuppressive network for unlimited immune escape potential [9].

The hallmark of cancer is extracellular acidification due to lactate accumulation. Previously, lactate was thought to be just a waste by-product of this metabolic process. However, emerging results show that lactate is one of the essential metabolites in TME; it acts as a fuel for mitochondrial metabolism, plays an indispensable role in shaping immune cell function, and cannot be overlooked in cancer immunotherapy [10,11,12]; it can regulate immune cell metabolism and inhibit the activation and proliferation of immune cells. As a signaling molecule, it plays an essential role in regulating the immune response of tumor cells, affecting immune surveillance and escape-related behaviors [10,11,12]. In addition, a novel function of lactate, protein lactylation, is newly proposed as a post-translational modification (PTM) of proteins to regulate gene expression [13]. Through histone lysine lactylation, lactate can promote the polarization of macrophages to an M2-like phenotype, thereby inhibiting immune responses within TME. Protein lactylation not only opens up a new field for studying proteins PTMs and guides a new direction for studying lactate in tumors, immunity, and other areas. However, a comprehensive understanding of the interactions between protein lactylation, metabolic reprogramming, and immunosuppression is lacking. Thus, further elucidation of the associations is necessary and pressing for more effective cancer therapy. We assessed the recent literature and briefly summarized the combined therapeutic strategies.

This review systematically elucidates the role of lactate as an immunosuppression molecule from the aspects of lactate metabolism and the effects of histone or non-histone lactylation on immune cells; it provides new ideas for further understanding protein lactylation in elucidating lactate regulation of cell metabolism and immune function. We explored the possibility of targeting potential targets in lactate metabolism for cancer treatment. Finally, we proposed a combined cancer therapy strategy that targeted glucose metabolism combined with immunotherapy.

## 2. Lactate Metabolism

Normal cells metabolize glucose through glycolysis, mitochondrial oxidative phosphorylation, and pentose phosphate pathways (Figure 1). Under aerobic conditions, pyruvate from glycolysis generates carbon dioxide and oxygen through oxidative phosphorylation (OXPHOS). In contrast, the glycolytic pathway is inhibited [14]. Under hypoxic conditions, glucose can be catabolized into two pyruvate molecules through glycolysis. Then, then, they produce two molecules of adenosine triphosphate (ATP) and nicotinamide adenine dinucleotide (NADH) at the same time [15]. During glycolysis, NADH and pyruvate are reduced to lactate and then excreted, and finally, each glucose molecule produces two molecules of ATP and two molecules of lactate without consuming oxygen [15].

Cancer cells, including glycolytic and oxidative cancer cells, are metabolically heterogeneous, depending on their intratumoral spatial localizations [16]. Specifically, glycolytic cancer cells are located away from the vasculature, and oxidative cancer cells are located near the vasculature. Lactate and protons are then co-exported through monocarboxylate transporters (MCTs) [17], producing extracellular lactate. Generally, lactate concentrations in normal serum range from 1.5 to 3 mM [18], while it can rise to 10–30 mM in tumor patients and even reach incredibly high levels (50 mM) in the inner tumor cores [19].

Several studies have indicated that high lactate concentrations can be transported into many cell types and metabolized as a fuel in TME [17,20,21]. Cancer cells far from blood vessels are under hypoxic conditions, mainly obtaining energy through glycolysis while producing excessive lactate, which is exported to the TME through MCT4 [22]. Cancer cells near blood vessels are normoxic and can oxidize lactate for ATP synthesis via MCT1. This lactate metabolic symbiosis occurs not only in cancer cells but also in other cells, such as cancer-associated fibroblasts (CAFs) and tumor-associated endothelial cells [23,24,25]. Similarly, Gladden LB et al. showed that the inhibition of MCT1 can disrupt the glycolysis process and inhibit glutathione synthesis in tumor cells, resulting in increased intracellular hydrogen peroxide content and mitochondrial damage, ultimately leading to cell death [26]. Therefore, lactate shuttling between cell populations is a prominent feature of the TME and is essential for tumorigenesis, progression, and metastasis.

## 3. Lactate-Rich Environment Mediates Immunosuppression

Excessive lactate facilitates the establishment of an immunosuppressive milieu that favors cancer cell growth and plays an essential role in shaping immune cell function [27]. Increasing studies focus on the interactions between lactate and multiple immune cells in TME to improve the efficacy of current antitumor immunotherapy. We will specifically discuss lactate-mediated antitumor immunity and focus on T lymphocytes, natural killer cells, T regulatory cells, dendritic cells (DCs), macrophages, and immunosuppressive myeloid-derived suppressor cells (MDSCs) (Figure 2).

### 3.1. T Cells

T cells sense extracellular lactate levels and cause intracellular signaling, regulation of cellular function, and homeostasis. Increased lactate concentration in TME interfere with T-cell function [28]. Excessive lactate inhibits T cell-mediated immune responses. When the pH is 6.0–6.5 in TME, activating effector CD8+ and CD4+ T cells are usually induced to an unresponsive state, with reduced cytolytic activity and cytokine production [24,29,30]. As mentioned, effector CD8+ and CD4+ T cells rely on glycolysis to translate Interferon γ (IFN-γ) and proliferation [24,30]. Similarly, studies have shown that neutralizing acidic TME and proton pump inhibitors can reverse the suppression of antitumor immunity and improve immunotherapy [31,32]. Another study further confirmed that lactate inhibited T cell receptor (TCR)-triggered IFN-γ, tumor necrosis factor α (TNF-α), and IL-2 production and impaired the function of cytotoxic T lymphocyte cells by inhibiting the phosphorylation of p38 signaling protein [33]. Lactate also induces T-cell apoptosis by reducing the levels of nicotinamide adenine dinucleotide (NAD(+)) [33,34].

Furthermore, lactate can regulate CD4+ T cell polarization and reduce the percentage of T helper 1 subsets by inducing SIRT1-mediated deacetylation/degradation of T-bet transcription factors [35]. Furthermore, lactate signaling in CD4+ T cells favors Th17 cell differentiation and suppresses T cell migration and trafficking [36,37].

More profound mechanistic studies have shown that lactate limits T cell proliferation through the NAD(H) redox state, reducing NAD+ to NADH under lactate-rich conditions, resulting in altered NAD+-dependent enzymatic reactions that reduce glycolytic intermediates required for proliferation [38].

### 3.2. Innate Lymphoid Cells

Group 2 innate lymphoid cells (ILC2s) and NK cells are part of the bigger family of innate lymphoid cells. Natural killer (NK) cells can directly secrete proinflammatory cytokines, perforin, and granzyme to exert antitumor effects. They are currently an important target in cancer immunotherapy. High lactate concentrations are reported to affect NK cytotoxic activity by reducing intracellular pH and inducing apoptosis of NK cells [39,40,41]. Furthermore, lactate inhibits the activation of nuclear factor-activated T cells (NFAT) in NK cells, thereby inhibiting the production of IFN-γ [42]. Elevated lactate levels not only directly constrain the cytolytic functions of natural killer (NK) cells but also indirectly restrain NK cells by increasing the number of MDSCs [43]. In addition, the researchers also found that the uptake of lactate by murine NK cells leads to intracellular acidification and accelerated apoptosis. Accumulation of lactic acid in tumor tissue inhibits the immune behavior of NK cells in multiple ways and accelerates tumor immune escape [42,44]. However, ILC2s constitute a relatively small proportion of immune cells in solid tumors, so the function of ILC2s remains uncertain [45]. Notably Wagner et al. [46]. Demonstrated that tumor-derived lactate significantly inhibited the function and activation of ILC2s, using melanoma as a model; it was also found that melanoma mice with reduced lactate production (LDHAlow) had delayed tumor growth and increased amounts of typical ILC2 compared to controls.

### 3.3. Regulatory T Cells

Not all immune cells respond negatively to lactate in TME. Regulatory T cells (Tregs), accounting for about 5% to 10% in human and mouse peripheral blood and lymphocytes, are potent immune system inhibitors responsible for maintaining immune homeostasis and preventing autoimmunity. The TME actively recruits and promotes the differentiation of Tregs by increasing the expression of Forkhead box protein P3 (FOXP3) and MCT1 [47,48]. High expression of FOXP3 reprograms Treg cell metabolism by inhibiting c-Myc and glycolysis, enhancing OXPHOS, and increasing NAD(+) oxidation, thus making Treg cells more adaptable in the low-glucose and high-lactate TME [30]. Additionally, MCT1-mediated lactate influx and intracellular lactate metabolism are essential for tumor-infiltrating Treg cells to sustain their suppressive activity, while high glucose levels dampen their function and stability [24]. In addition, lactic acid was shown to be critical for tumor-infiltrating Treg proliferation and function [24]. The authors observed Treg-specific MCT1-deficient mice and found that Treg cell survival and function were not significantly affected despite the lack of lactate uptake. However, injection of B16 melanoma in MCT1-deficient mice resulted in slower tumor growth and more prolonged survival of the mice [24]. Similar to the previous results, another article has also proposed that a large amount of lactate secreted by glycolytic tumors can achieve tumor immune evasion. They maintain the immunosuppressive function of Treg cells and inhibit the function of effector T cells [49].

In summary, the large number of Tregs in TME is now considered the “culprit” for tumor cells evading immune surveillance. Clearing the aggregated Tregs in the TME and inhibiting the differentiation and expansion of Tregs are also promising development directions in tumor immunotherapy. However, there is little research on Treg’s molecular mechanism of lactate regulation. Recent studies have shown that protein lactylation is important for lactate to function. Therefore, lactylation will primarily become a critical entry point for future research on the regulation mechanism of immune cells such as Treg.

### 3.4. Dendritic Cells

DCs are the most important antigen-presenting cells, which can efficiently ingest, process, and transmit antigen information to CD8+ T cells. Lactate inhibits the differentiation of monocytes into DCs, activation, and antigen degradation and inactivates cytokines secreted by differentiated DCs, leading to differentiation into resistant DCs [50,51]. Lactate also inhibits DC differentiation through the induction of IL-10 production with concomitant loss of IL-12 in response to Toll-like receptor (TLR) stimulation [50]. In line with the above findings, Gottfried et al. [51] found that lactate, during the differentiation of DCs in vitro, induced a phenotype that is down-regulation of IL-12 secretion and CD1a expression. This phenotype is consistent with tumor-associated dendritic cells (TADCs) generated in melanoma and prostate carcinosarcoma. Moreover, the blockade of lactate production in melanoma co-cultures restored the phenotype of TADCs.

Similarly, lactate-mediated activation of G protein-coupled receptor 81(GPR81), G protein-coupled cell surface lactate receptor, in DCs was found to abolish antigen presentation, secretion of proinflammatory cytokines IL-6 and IL-12, and T cell function and promotes tumor progression in a mouse model [52].

Simultaneously, the study revealed that lactate enhanced tryptophan catabolism and kynurenine production in plasmacytoid dendritic cells (pDCs) and immunosuppressive FoxP3+CD4+ regulatory T cells, which results in the immunosuppression of the TME [53,54]. Furthermore, excess lactate inhibits IFN-α production by pDCs through two distinct mechanisms. On the one hand, the binding of lactate to the GPR81 receptor on the surface of pDCs leads to intracellular calcium mobilization and inhibits IFN-γ production [53]. On the other hand, high extracellular lactate concentration inhibits the export of lactate from DCs, which leads to the accumulation of lactic acid in the cytoplasm of pDCs and affects the process of glycolysis [55]. Glycolysis is one of the leading energy supply pathways for pDCs to induce IFN-γ production in response to TLR stimulation [56]. Therefore, lactate inhibits the glycolysis process and the output of IFN-γ by affecting the energy production pathway of pDCs.

Thus, the lactate-induced tolerance phenotype of tumor-infiltrating DCs indirectly affects T lymphocyte priming and promotes immunosuppressive cytokine profiles and Treg expansion, enhancing tumor immune escape.

### 3.5. Macrophages

Macrophages’ phenotype depends on the anatomic region where they are present. They can switch their character according to their environment; it is reported that tumor-derived lactate directly guides cancer-associated macrophages to become M2-like polarised cells, promoting tumor growth in the TME [57]. Mechanically, the extracellular signal-regulated kinase (ERK)/transcription 3 (STAT3) activation, the activator of STAT3 signaling pathway and stimulated expression of vascular endothelial growth factor (VEGF) and arginase-1 (ARG1), and stabilization of HIF-1a contributes to the lactate-induced M2 macrophages polarization and its pro-tumorigenic effects in breast cancer [58,59]. Moreover, tumor-derived lactate can induce tumor-associated macrophage (TAM) polarization to the M2 immunosuppressive phenotype by binding to the lactate-sensitive receptor-G protein-coupled receptor 132(GPR132) [57,58]. Taken together, it is conceivable that the lactate-rich landscape in TME drives the re-education of TAMs to an M2 phenotype. In addition, M2 macrophages secrete immunosuppressive cytokines that inhibit tumor-infiltrating lymphocytes (TILs) cytotoxicity and promote the differentiation of Tregs [60].

Lactate affects macrophage metabolical reprograms and immunomodulatory effects. Mainly, it promotes polarization changes that negatively affect tumor immune behavior. Compared to the glycolytic metabolism in M1 macrophages, M2 TAMs rely on OXPHOS to meet their bioenergetic demands. This trait may additionally support the metabolic symbiosis between highly glycolytic triple-negative breast cancer (TNBC) tumor cells and M2 TAMs [61]. In addition, elevated extracellular lactate levels prevent the excretion of lactate produced in macrophage precursor monocytes, thereby promoting a negative feedback mechanism for glycolysis and TNF-α release [62]. In TLR-activated monocytes, tumor-secreted lactate enhances IL-23/IL-17 transcription, thereby polarizing the immune response to a pro-tumor Th17 subset while suppressing antitumor Th1 responses [63,64]. Therefore, lactate adversely affects macrophage function, polarization, and precursor monocytes.

### 3.6. Myeloid-Derived Myeloid Suppressor Cells

MDSCs are immunosuppressive immune cells that are the most prominent cell population. They exert broad immunosuppressive functions and have a significant ability to suppress T cell responses, such as restricting T cell function, proliferation, and TCR signaling and promoting differentiation of Tregs [65]. Studies have shown that glycolytic gene expression profiles correlate with MDSCs gene expression, and both are associated with reduced survival [66]. Increased glycolysis and elevated lactate concentrations induce MDSC development and immunosuppression. Thereby, it promoted granulocyte stimulating factor (G-CSF) and granulocyte-macrophage stimulating factor (GM-CSF) in a mouse model of TNBC [66]. In addition, lactate can increase the expansion frequency of MDSCs, inhibit the function of NK cells and limit the activity of innate immune effectors; it indicates that lactate can establish a tumor immunosuppressive microenvironment to promote the occurrence and development of tumors by regulating MSDCs [67]. As regulators in tumor immunity, MDSCs play an essential role in inhibiting the functions of different immune cells involved in the process of immune escape. Targeting the regulation of MDSCs and reducing their inhibitory effect has become one of the effective methods for tumor therapy.

## 4. Lactylation Mediates Immunosuppression

The latest study found that lactate acts as an epigenetic regulatory molecule and can regulate the expression of related genes through the epigenetic modification of histone lactylation [10]. The imbalance between glycolysis and the tricarboxylic acid cycle (TCA) caused by cellular metabolic reprogramming is essential in increased histone lactylation [68]. In addition, histone lactylation is regulated by other critical enzymes of cellular metabolism. There is evidence that p300 enzymes are involved in the enzymatic process of histone lactylation [69]. The p300 enzyme, a histone acetyltransferase, can transfer the acetyl group of acetyl-CoA to lysine residues on histones to form histone acetylation.

Interestingly, the lack of acetylation in promoter regions enriched for histone lactylation modification indicates that the dual roles of p300 enzymatic acetylation and lactation have different temporal and spatial dynamics in regulating gene expression [69]. Consistent with previous studies, Zhang et al. identified 26 and 16 histone Kla sites, including H3, H4, H2A, and H2B. Lactation of H3 and H4 depends on the p53 gene and is mediated by p300 [70]. In addition, some studies have demonstrated that the lactation of histone or other protein lysine residues may have a non-enzymatic reaction [71].

Lactate, a metabolic by-product, significantly inhibits the function of immune cells in the TME, and lactylation reveals the underlying mechanism by which lactate regulates cellular metabolism and function. Zhang et al. [10] identified that lipopolysaccharide (LPS)-activated M1 macrophages enhanced glycolytic metabolism and increased lactate concentration. Moreover, increasing histone lactylation resulted in increased Arg1 and other wound healing–associated gene expression, suggesting a shift to the immunosuppressive M2 macrophage phenotype (Figure 3).

They also found that macrophages are unable to produce lactate. LPS increases the expression of inflammation-related genes. However, the presentation of histone lysine lactation and homeostatic genes could not be upregulated, which suggests that histone lactylation is essential in bacteria. Infected M1 macrophages play an important role in homeostasis regulation. Another study [72] showed that macrophage-intrinsic B cell receptor-associated protein (BCAP) has roles in intestinal inflammation and tissue repair and that lack of BCAP results in the failure of macrophage transition from an inflammatory state to a repair state. While histone lactylation plays an essential role in BCAP-promoted macrophage transition, BCAP-deficient macrophages blunt their repair transition due to impaired lactate production resulting in reduced histone lactylation and tissue repair gene expression.

Similarly, a recent study [73] showed that the lactate content in the medium of transforming growth factor (TGF-β) induced lung fibroblasts and bronchoalveolar lavage fluid (BALF) of mice with TGF-β or bleomycin-induced pulmonary fibrosis A significant increase. Lactate in culture medium and BALF promotes pulmonary fibrosis gene expression by upregulating histone lactate modification in macrophages mediated by p300. Lactation may play a similar role in Tregs to induce or enhance the presentation of immunosuppressive genes [24,49]. Zappasodi et al. revealed that glycolytic activity in tumors and Treg cells blocks CTLA-4 and those glycolytici tumors might be more sensitive than glycolysis-high tumors to anti-CTLA-4 treatment [49]. Therefore, these studies suggest that the lactate molecule plays an essential role in regulating immune homeostasis through epigenetic modification of histone lactylation.

In addition, histone lactylation plays an essential role in tumorigenesis and the maintenance of tissue homeostasis [74,75,76]. In addition, lactate induces substantial amounts of α-ketoglutarate and is ultimately associated with widespread epigenome reprogramming in pancreatic ductal adenocarcinoma by lactylation [77]. Yu et al. revealed that histone lactylation enhances tumorigenesis by promoting transcription of YTH N6-methyladenosine RNA-binding protein 2 (YTHDF2), which recognizes m6A modification sites in the mRNA of two tumor suppressor genes, period1(PER1), and TP53; it promotes their degradation in ocular melanoma [78]. Choi et al. recently discovered that the lack of NAD+-dependent histone deacetylase sirtuin 6 (SIRT6) enriches tumor-propagating cells (TPCs) by improving lactate production, which leads to a more aggressive tumorigenic phenotype in squamous cell carcinoma (SCC) [79]. Another study reported that histone Kla is essential in the oncogenic process. For example, the continued increase in the number of lactic acid bacteria in gastric cancer patients suggests that these bacteria may contribute to gastric cancer by providing lactate, and lactylation may be implicated in this process [80].

## 5. Crosstalk between Lactylation and Other Acylations

Chemical modifications of proteins are also referred to as PTMs. These modifications, which include lysine lacylation, can regulate protein activity, turnover, localization, and dynamic interactions with cellular molecules, such as other proteins, nucleic acids, lipids, and connectors [81,82]. There is potential crosstalk among lysine acylations since their corresponding metabolic sources are intertwined. Therefore, it is essential to note that metabolic pathways are interconnected and coordinately regulated when explaining the link between lacylation and other PTMs [83].

It is reported that many proteins contain at least one type of regulatory PTM. Most proteins interacted with other proteins to some extent, suggesting that Crosstalk is ubiquitous among PTMs of different proteins [84]. Emerging studies indicate acetylation and lactylation exist in diverse non-histone proteins [85,86]. Histone lactylation and acetylation are essential processes linking metabolism and epigenetics. Gli-like transcription factor 1(Gli1) increases glycolysis by opening glycolytic genes and closing somatic genes, which produce more lactate and acetyl-CoA [87]. Subsequently, improved acetylation and lactylation promote cellular reprogramming, which induces senescence in pluripotent stem cells [87]. ChIP-seq indicated the enrichment of pan-Kla and H3K18 la in the promoters at the Oct4, Sall4, and Myc loci, which exhibited patterns parallel to those of H3K27Ac, coordinated with histone acetylation and lactylation in glycolysis-dependent processes such as pluripotency acquisition. A recent study in CD4+ Th1 cells demonstrated that glycolysis is essential for acetyl-CoA levels during T cell differentiation [88]. Lactate dehydrogenase (LDHA) activity is mechanically important for acetyl-CoA production, enhancing H3K9Ac and H3K27Ac levels and IFN-g expression in Th1 cells [88]. The stories of acetyl-CoA and lactate are altered simultaneously in many cells that depend on glycolysis.

Moreover, intracellular lactate can cause epigenetic modifications via NAD(+)-independent histone deacetylase inhibition and histone lysine residue lactylation in macrophages [10,89]. Yang et al. recently reported that lactate enhances acetylation and lactylation of high mobility group protein B1 (HMGB1) and its release from macrophages via exosomes; it induces vascular endothelial cell injury by decreasing the steady state and promoting vascular permeability [90]. Mechanically, intracellular metabolic crosstalk is associated with the ability of epigenetic writers and eraser enzymes, which are regulated by histone acetylation and lactylation [91].

In addition to acetylation, lactylation also interacts with other acylation modifications. In LPS-activated macrophages, succinylation of pyruvate kinase two at lysine residue K311 impairs glycolytic activity; it induces nuclear translocation, promoting the transcription of hypoxia-inducible factor (HIF)-dependent genes and producing IL-1β [92,93]. AYu et al. [78] have revealed the oncogenic role of histone lactylation; it provides a novel association between histone and RNA modifications and furnishes a new sight for epigenetic regulation in carcinogenesis. Lactylation might crosstalk with other PTMs to maintain the homeostasis of TME and is carcinogenic. The latest research [94] reveals that lactate modification regulates the immunosuppressive function of tumor-infiltrating myeloid cells (TIMs) to mediate tumor immune escape. The study found that methyltransferase-like 3 (METTL3) expression was upregulated in TIMs and associated with poor prognosis. Mechanically, lactate promotes the transcription of Mettl3 in TIMs through histone H3K18 lactate modification. What is more, lactylation can directly occur in the “CCCH” zinc finger domain (ZFD) of METTL3 protein. ZFD plays the target recognition domain (TRD), enhancing METTL3 binding and catalyzing m6A modification of target RNA and enhancing the expression of downstream immunosuppressive effector molecules IL-6, IL-10, iNOS, etc. through the METTL3-JAK1-STAT3 regulatory axis. Therefore, it is necessary to find novel therapeutic targets and strategies based on the crosstalk between lactylation and other acylations for tumor therapy.

## 6. Advances in Targeting Lactate-Lactylation Combined with Immune Checkpoint Inhibitors

Lactate plays an essential role in tumor metabolic reprogramming; it is considered a necessary tool for engineering the TME. Abnormal lactate concentrations have been shown to affect the differentiation, metabolism, and function of tumor-infiltrating immune cells through multiple pathways. Recognition of the abundance and immunomodulatory impacts of lactate and lactylation has led to the development of novel targets to improve cancer immunotherapies. Moreover, it will be essential to understand how inhibition of lactate and lactylation will interact with a range of immunotherapies. In tumors, metabolites in the TME have a substantial inhibitory effect on immune cells, especially antitumor T cells. Therefore, the metabolic treatments that modulate glucose metabolism to improve the TME could be attractive adjuvants to be used in combination with ICIs. Further understanding of how these toxic metabolites can be limited to altering immune cell function will help the field utilize the proper immunotherapy to achieve maximal efficacy.

Many strategies combine to alter tumors’ metabolic landscape to enhance tumor immunotherapy’s efficacy; it has been reported that in tumors with low glucose and high glycolysis metabolism, such as tumors with high MYC gene expression and liver metastases, Tregs cells actively take up lactate, resulting in increased expression of PD-1 (Figure 4). In this case, PD-1 inhibitor treatment will activate PD-1+ Treg, further inhibiting PD-1+ CD8+ T cells and ultimately leading to treatment failure [95]. Therefore, lactate may be an effective checkpoint to determine Treg function in the hyperglycolytic TME. Reducing lactate in cancer cells may be related to tumor immunotherapy, such as PD-1 inhibitors, producing synergy. Lactate can be inhibited by targeted inhibition of LDH or by inhibiting other glycolytic enzymes.

LDH levels in blood and TME correlate with poor prognosis in cancer patients and can be used to determine tumor staging in melanoma [96]. For melanoma patients, high LDH levels predict poor response to anti–PD-1 immunotherapy [97,98]. In a mouse melanoma model, it has been reported that an increase in IFN-γ and granzyme B production in NK cells and CD8+ T cells can be observed, and an increase in PD-1 Antitumor immune responses to immune checkpoint inhibitors by blocking LDH-A [99]. To date, glycolysis inhibitors are still in the preclinical stage, so their effects in humans are unclear. Another study demonstrated that inhibition of patient-derived and B16 melanoma LDHA with the inhibitor GSK2837808A could enhance T cell function in vitro and in vivo and enhance adoptive cell therapy [100].

Similarly, Jian Gu et [101]. revealed that lactate catalyzes the lactation of lysine at position K72 of MOESIN protein, improves the interaction of MOESIN with TGF-β receptor I and downstream SMAD3 signaling, and regulates the generation of Treg cells. Combination therapy of LDH inhibitor and anti-PD-1 enhances its antitumor effect. In patients with hepatocellular carcinoma who responded to anti-PD-1 treatment, the level of MOESIN protein lactylation in Treg cells was lower than that of non-responders. Moreover, Metformin inhibits tumor cell oxygen consumption and reduces intratumoral hypoxia. Metformin was first used to treat type 2 diabetes in the late 1950s and, in 2022 remains the first-choice drug for improving insulin sensitivity. Pre-clinical and clinical studies demonstrated that metformin could treat a variety of diseases for its anti-bacterial, -viral, -malaria efficacy. What is more, metformin may protect against neurodegenerative diseases and direct actions on neuronal stem cells via suppression of proinflammatory pathways, and protection of mitochondria and vascular function. A combination of Metformin with anti-PD-1 improves T-cell function and tumor clearance in mice with melanoma [101]. Favorable outcomes were also observed in melanoma patients receiving Metformin in combination with ICI [102]. Similarly, non-small cell lung cancer patients receiving concurrent Metformin and ICIs showed a higher response rate and overall survival [103].

In addition, lactate can also be decreased within the TME by targeting its export. Lactic acid is transported via MCTs [104]. MCT1 has the highest affinity for lactate and can import and export lactate based on the concentration gradient of a substrate, and protons MCT4 are more heavily expressed by highly glycolytic tissues, including tumor cells. While it is also a two-way transporter, it is mainly involved in the export of lactate [104]. Although many small molecule inhibitors of MCT1 and MCT4 have been developed in preclinical studies, such as 7ACC2, AR-C155858, syrosingopine, and AZD3965, only AstraZeneca’s AZD3965 compound is currently being tested for use in humans (ClinicalTrials.gov NCT01791595). Preclinical trials have shown that combined therapy with the MCT1 inhibitor AZD3965 and anti-PD-1 treatment reduces the lactic acid secretion into the TME and the infiltration of exhaustive PD-1+ Tim-3+ T cells in solid tumors and improves antitumor immunity [105]. In light of this, it helped inhibit tumor growth and combined with anti–PD-1 therapy by knocking out MCT1 specifically on Tregs; it suggests that pharmacologic inhibition of MCT1 may play a dual role in inhibiting lactate secretion [100]. In hepatocellular carcinoma (HCC) patients, overexpression of MCT4 is positively associated with poor patient prognosis. Overexpression of MCT4 results in suppressed CD8+ T cell recruitment and reduced activity in a mouse model of HCC [106]. By using an MCT4 inhibitor, the acidification process of TME can be effectively blocked, and the expression of chemokine (CXC motif) ligands CXCL9 and CXCL10 can be induced through the ROS/NF-κB signaling pathway. At the same time, in tumor mouse models, inhibition of MCT4 can significantly enhance the therapeutic effect of PD-1 immune checkpoint inhibitors [107], which indicates that the combined use of MCT4 inhibitors may provide opportunities for immunotherapy-resistant HCC patients.

Lactate induces an acidic environment. Therefore, raising the pH of the TME can improve cancer therapy. Bicarbonate has been used by various methods [108]. Bicarbonate administration has been reported to control melanoma growth, increase CD8+ T cell infiltration and NK and B cell activation, and improve anti-CTLA4 and anti–PD-1 therapy and adoptive cell therapy in B16 melanoma–bearing mice [109,110]. Future research on the combination of metabolic modulation therapy with ICI is warranted.

## 7. Conclusions

In the last decade, cancer immunotherapy has become one of the most promising types of treatment. However, currently available immunotherapies focus only on restoring or enhancing isolated components of the immune system, often directed at a single immune cell type. For example, over 2000 clinical trials are investigating PD-1/PD-L1-targeted drugs [110], yet only a minority of patients will respond to these agents [111]. Therefore, exploring combination therapy is a necessary and feasible research direction.

Lactate is essential in the TME and was previously considered a metabolic waste. However, recently, emerging studies have demonstrated that lactate can play a significant role in promoting tumor progression, such as being metabolized as a fuel substrate, promoting tumor invasion and metastasis, promoting tumor angiogenesis, and mediating immune suppression. Surprisingly, a single metabolite can have a powerful effect on immune cell function; it is now recognized as an effective target for cancer therapy. Multiple studies have shown that drugs targeting tumor aerobic glycolysis inhibit T cell function and promote immunosuppression [112]. Progress has been made in novel inhibitors of the aerobic glycolysis pathway and related proteins such as lactate production and transport, such as MCTs and LDH. The combination of tumor metabolic reprogramming and tumor immunity may become an essential direction of tumor immunotherapy.

Zhang et al. discovered that histone lactylation is a new PTM that takes lactate as a substrate [10]. The discovery of histone lactylation and its impact on macrophage biology is a blueprint for understanding how lactic acid changes other cell types, unlocking the mysteries of the Warburg effect and understanding its implications for human disease. Unfortunately, many important scientific questions still need to be addressed. For example, it is unclear whether lactylation occurs at amino acid residues other than lysine. Secondly, lactate-mediated lactylation is more broadly involved, given the broad functions of Lactate in the TME. Thirdly, the enzymes producing lactyl-CoA and mechanisms by which writer, eraser regulate histone Kla, and reader enzymes remain to be studied. Finally, whether lactylation crosstalk with other PTMs, such as acetylation and methylation in different diseases or disease processes, influences the prognosis.

Evidence suggests that a cellular metabolic metabolite, lactate, can remodel the TME, regulate cellular metabolic reprogramming, and modulate antitumor immunity. Protein lactylation is a crucial way for lactate to function, which provides new ideas for studying the molecular regulation mechanism downstream of the Warburg-like effect. Given the extensive functions of lactate in the TME, is lactylation based on lactate more extensive? It is indispensable to analyze the regulatory mechanism and functional exploration of lactylation, which is of great significance for understanding the pathogenesis of the disease and discovering new therapeutic targets and strategies. This review detailed and systematically summarized the remodeling effect on TME and the immunosuppressive effect mediated by lactate-lactylation. Through this article, we find the metabolic rewiring of tumor and immune cells regulates tumor progression by shaping the epigenome in the TME.

What is more, varieties of strategies targeting metabolic pathways have been developed to enhance immunotherapy. All in all, lactate-lactylation hands tumor metabolic reprogramming and tumor immunity. This will become an essential direction of tumor immunotherapy. Hence, further investigating the role of histone lactylation in modulating the activity of antitumor immunotherapy and further cancer development will lead to intriguing findings.

## Figures and Tables

**Figure 1 ijms-23-11943-f001:**
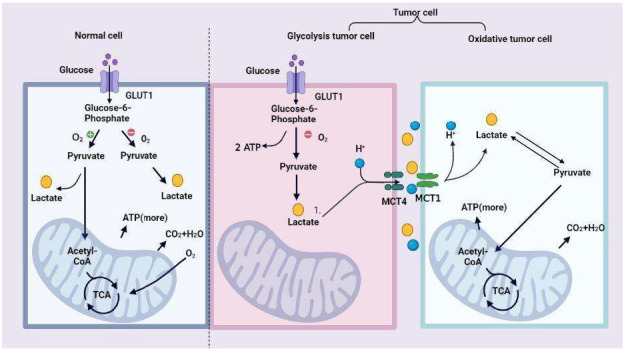
Regulation of lactate metabolism progress in normal and cancer cells. Glucose metabolism mainly contains glycolysis and the TCA cycle in the mitochondrion. With sufficient oxygen, normal cells produce energy mainly through the TCA cycle. Under hypoxic conditions, large amounts of lactate are produced. Glycolysis tumor cells produce large amounts of lactate, which transport into TME through the MCT4. Then, lactate can be transported into oxidative tumor cells based on MCT1 as fuel and produce energy by OXPHOS. Abbreviation: ATP, adenosine triphosphate; TCA cycle, tricarboxylic acid cycle; TME, tumor microenvironment; MCT1, monocarboxylate transporter 1; MCT4, monocarboxylate transporter 4; OXPHOS, oxidative phosphorylation, GLUT1, glucose transporter type 1.

**Figure 2 ijms-23-11943-f002:**
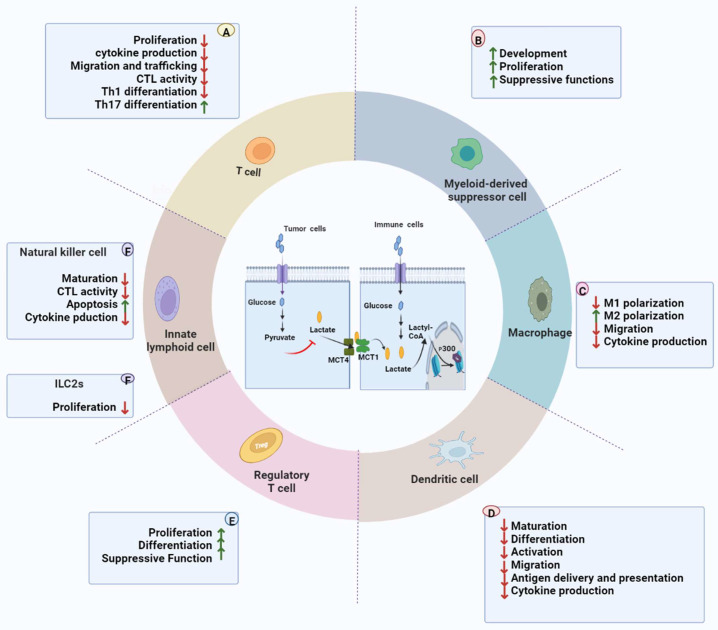
Lactate-lactylation forms an inhibitory regulatory network on the immune system in the TME. In the TME, tumor cells consume most nutrients and secrete excessive lactate, resulting in acidosis, angiogenesis, and immunosuppression. Lactate also modulates the metabolism of innate and adaptive immune cells by inhibiting the functions of CD8+ T cells, natural killer (NK) cells, natural killer T (NKT) cells, dendritic cells, and macrophages. By contrast, lactate favors regulatory T (Treg) cells and Myeloid-Derived Myeloid Suppressor Cells (MDSCs), sustaining their immunosuppressive functions in the acidic environment. Summarily, lactate plays a pro-oncogenic role in TME. Abbreviation: ILC2s, group 2 innate lymphoid cells; CTL, cytotoxic T lymphocytes, MCT1, monocarboxylate transporter 1; MCT4, monocarboxylate transporter 4.

**Figure 3 ijms-23-11943-f003:**
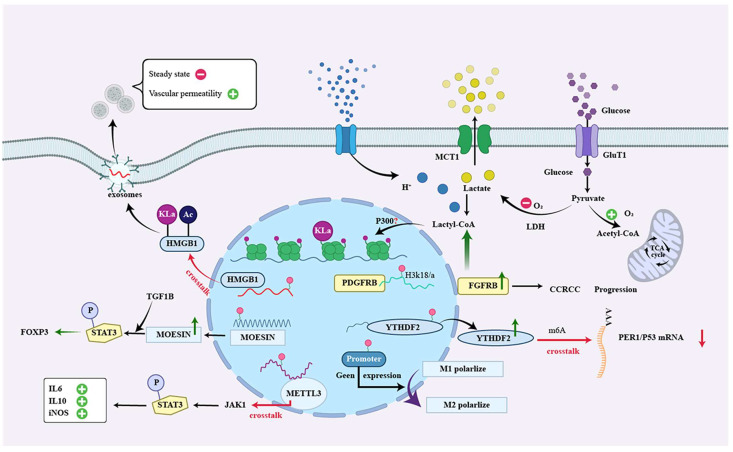
Lactate acts as a signaling molecule to affect gene transcription and immune evasion via histones and non-histone lysine lactylation. Lactate promotes lactylation and acetylation of HMGB1 and its release from macrophages via exosomes; it induces vascular endothelial cell injury by decreasing the steady state and promoting vascular permeability. Lactation of lysine at position K72 of MOESIN protein improves the interaction of MOESIN with TGF-β receptor I and downstream SMAD3 signaling and regulates the generation of Treg cells; lactate promotes the transcription of METTL3 in TIMs in the form of histone H3K18 lactate modification. Moreover, lactylation can directly occur in METTL3 protein, which enhances METTL3 binding and catalyzes m6A modification of target RNA; it enhances the expression of downstream immunosuppressive effector molecules IL-6, IL-10, and iNOS. Histone lactylation in macrophages promotes a shift to the immunosuppressive M2 macrophage phenotype. Histone lactylation promotes tumorigenesis by facilitating the transcription of YTH N6-methyladenosine RNA-binding protein 2 (YTHDF2), which recognizes the m6A modification site in the mRNA of two tumor suppressor genes, PER1, and TP53; it promotes their degradation in ocular melanoma; Lactylation of PDGFRB in the histone H3K18 is essential in the oncogenic process. Abbreviation: Kla, histone lysine lactylation; MCT1, monocarboxylate transporter 1; GLUT1, glucose transporter type 1; Ac, Acetylation; METTL3,methyltransferase-like 3; TCA cycle, tricarboxylic acid cycle; YTHDF2,YTH N6-methyladenosine RNA-binding protein 2; PER1, period1; CCRCC, Clear Cell Renal Cell Carcinoma; LDH, lactate dehydrogenase; TGF1B, TGF-β receptor I; STAT3,transcription 3; FOXP3, forkhead box protein p3; JAK1, Janus kinase 1; HMGB1, high mobility group protein B1.

**Figure 4 ijms-23-11943-f004:**
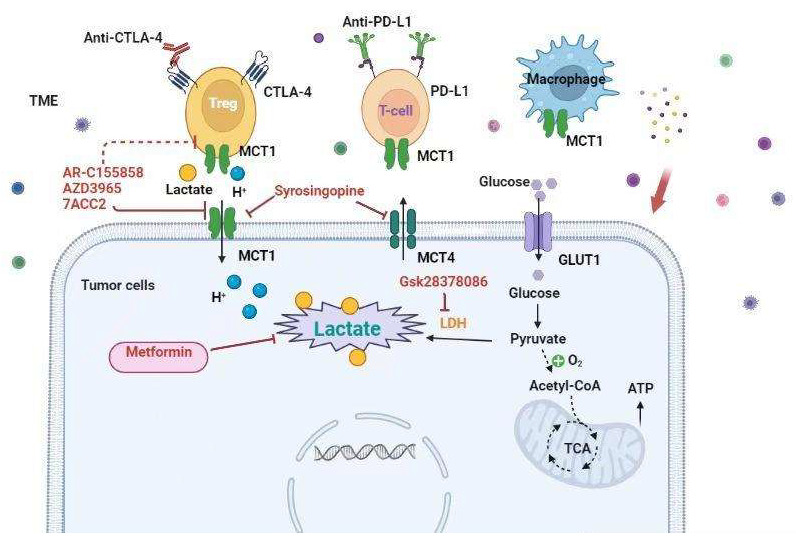
Strategies to target lactate biogenesis and acidosis to enhance immunotherapy response. Combined immunotherapy with drugs targeting lactate production and lactate transporter can enhance the therapeutic effect of immune checkpoint inhibitors.

## Data Availability

Not applicable.

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
