# Peer review of "Lactate-Lactylation Hands between Metabolic Reprogramming and Immunosuppression"

_ijms, 2022, doi:10.3390/ijms231911943_

Round 1

Reviewer 1 Report

This is a well-written and comprehensive review article focusing on the role of lactate in the tumor microenvironment. The article is interesting, however there are some minor comments that would increase the quality of the article.

An important article showing the impact of lactate on ILC2s is missing (Wagner et al., 2020, https://doi.org/10.1016/j.celrep.2020.01.103). ILC2s (together with NK cells, which have been described) are part of the bigger family of innate lymphoid cells. Those two cell types should be discussed together in the same paragraph. 

Lines 49-51: Please rewrite the sentence. Sounds like myeloid cells can gain unlimited immune escape potential, which doesn't make sense. 

Line 81: Please rewrite the sentence. It is confusing.

Line 87: Please include examples of glycolytic and oxidative cancer cells.

Line 90: Should read "lactate" rather than "Lactate". That applies to the whole manuscript. 

Line 218: Should be "promoting tumor growth"??? M2 macrophages stimulate tumor growth and progression. 

Line 372- 374: Please replace "On the other hand" since it has been written twice.

Line 437: Please elaborate more on Metformin and its role. 

Line 446: Space is missing (i.e. tonsMCT4).

Line 466: Should read "pH".

Line 512: Please rewrite the sentence. It doesn't make sense. "This review has a ..."

Line 518: Please replace "In a word".

Last but not least, some figures lack explanation of abbreviations. 

Author Response

Professor Editorial Office

Editor-in-Chief

Manuscript title: Lactate-lactylation Hands between Metabolic Reprogramming and Immunosuppression

Submission ID: ijms-1939060

Dear editors and reviewers :
Thank you so much for your arranging a timely review for our manuscript. We are excited to receive the letter from your editorial office. We would like to thank all members of the editor team and the peer reviewers for their helpful suggestions and remarks. We would like thank you again for the chance to submit a revision version. As soon as we received the last decision letter, we held a group meeting to address all of the critiques mentioned, with particular focus on the issues that need to be improved.

To the best of our knowledge, we did consider all topics that required a further attention.Any revisions of the manuscript were marked up using the“Track Changes” function with MS Word. We are confident that the present version of the manuscript is far more stringent and straightforward. Furthermore, an English language editor has reviewed the revised manuscript and corrected any grammar errors. We wish to refer to the comments systematically, and detailed corrections are listed below point by point.

Response to Reviewers

  1. An important article showing the impact of lactate on ILC2s is missing (Wagner et al., 2020, https://doi.org/10.1016/j.celrep.2020.01.103). ILC2s (together with NK cells, which have been described) are part of the bigger family of innate lymphoid cells. Those two cell types should be discussed together in the same paragraph.

Response: Thank you for the great suggestions. According to the reviewer’s suggestion, we have added the content of the impact of lactate on ILC2s in lines 161-166, in the revised manuscript.

  1. Lines 49-51: Please rewrite the sentence. Sounds like myeloid cells can gain unlimited immune escape potential, which doesn't make sense.

Response: We are very sorry for our poor English expression. We have rewritten this sentence in line 50-51.

  1. Line 81: Please rewrite the sentence. It is confusing.

Response: We are very sorry for our poor English expression. We have rewritten this sentence in line 81-83.

  1. Line 87: Please include examples of glycolytic and oxidative cancer cells.

Response: Thanks for your comments again and we have added the content in line 92-93.

  1. Line 90: Should read "lactate" rather than "Lactate". That applies to the whole manuscript.

Response: Thank you for your suggestion. Indeed, there is a section of "Lactate" should be replaced with "lactate" through the whole manuscript. We have checked carefully and revised.

  1. Line 218: Should be "promoting tumor growth"??? M2 macrophages stimulate tumor growth and progression.

Response: Thank you for helping us find the vague description in the manuscript. We are very sorry for our incorrect writing, and we have revised carefully again and rewritten it in line 235.

  1. Line 372- 374: Please replace "On the other hand" since it has been written twice.

Response: We are very sorry for our poor English expression, and we have revised them in line 392-393.

  1. Line 437: Please elaborate more on Metformin and its role.

Response: That is really an excellent comment. We do agree with the reviewer’s suggestion.  We have added this section in line 459-465.

  1. Line 446: Space is missing (i.e. tonsMCT4).Line 466: Should read "pH".

Response: Thank you for helping us find the irregular writing in this manuscript, and we have revised carefully again and revised them in line 473 and line 494

  1. Line 512: Please rewrite the sentence. It doesn't make sense. "This review has a ..."

Response: We are very sorry for our poor English expression. Indeed, this sentence is similar to the following sentence, so we deleted it directly.

  1. Line 518: Please replace "In a word".Last but not least, some figures lack explanation of abbreviations.

Response: hank you for helping us find the vague description in the manuscript. We are very sorry for our incorrect writing, and we have revised carefully again and rewritten it in line 550. And we have checked carefully again and added the explanation of abbreviations in revised manuscript.

Finally, we thank you again for your help in improving the quality of our manuscript. We tried our best to improve our manuscripts to match your journals criterion. And all the revisions were marked as bright red in this version.

We do earnestly thank to the editors/reviewers’ work and their helpful comments and suggestions and we would like address our deeply gratitude.

Sincerely,

Correspondence to:

Yang Xiang, MD. Professor, Peking Union Medical College Hospital, Chinese Academy of Medical Sciences, Peking Union Medical College, No.1 Shuai fu yuan Wang fu jing, Dong cheng District, Beijing, 100730, People’s Republic of China.

Email: [email protected].

Tel: 0086-01065296068

Fax: 0086-01065296218

Reviewer 2 Report

Comments for the author:

The review article submitted by Chen et al. discusses recent information around lactate-lactylation and immunosuppression. The manuscript nicely summarizes the role of lactate as an immunosuppressive molecule and the possibility of targeting the lactate pathway for cancer treatment. There are some minor concerns before its final publication.

1.       None of the Figures have been referenced/ called out in the text. They are all missing.

2.       Modify Figure 2 as some of the text are not visible with the background color.

3.       Page 9 line 366 mentions “It provides a novel between …” - Please rephrase the sentence as it is missing some context grammatically.

4.       In the Abbreviation table –

a.       please bring everything to lower case. Maintain consistency.

b.       G-CSF and GM-CSF have same definition. Please correct it.

Author Response

Professor Editorial Office

Editor-in-Chief

Manuscript title: Lactate-lactylation Hands between Metabolic Reprogramming and Immunosuppression

Submission ID: ijms-1939060

Dear editors and reviewers :
Thank you so much for your arranging a timely review for our manuscript. We are excited to receive the letter from your editorial office. We would like to thank all members of the editor team and the peer reviewers for their helpful suggestions and remarks. We would like thank you again for the chance to submit a revision version. As soon as we received the last decision letter, we held a group meeting to address all of the critiques mentioned, with particular focus on the issues that need to be improved.

To the best of our knowledge, we did consider all topics that required a further attention.Any revisions of the manuscript were marked up using the“Track Changes” function with MS Word. We are confident that the present version of the manuscript is far more stringent and straightforward. Furthermore, an English language editor has reviewed the revised manuscript and corrected any grammar errors. We wish to refer to the comments systematically, and detailed corrections are listed below point by point.

Response to reviewer #2

  1. None of the Figures have been referenced/ called out in the text. They are all missing.

Response: Thank you for your suggestion. I would like to declare that all figures in this review are original, and there are no cited references.

  1. Modify Figure 2 as some of the text are not visible with the background color.

Response: Thank you for your suggestion. That is really an excellent comment. We have adjusted the background color of Figure 2 to make the text stand out even more.

  1. Page 9 line 366 mentions “It provides a novel between …” - Please rephrase the sentence as it is missing some context grammatically.

Response: Thank you for helping us find the vague description in the manuscript. We are very sorry for our incorrect writing, and we have it in page9, line 386

  1. In the Abbreviation table: please bring everything to lower case. Maintain consistency. b. G-CSF and GM-CSF have same definition. Please correct it.

Response: Thank you for your suggestion. We checked the abbreviation table and revised them in the revision manuscript.

Finally, we thank you again for your help in improving the quality of our manuscript. We tried our best to improve our manuscripts to match your journals criterion. And all the revisions were marked as bright red in this version.

We do earnestly thank to the editors/reviewers’ work and their helpful comments and suggestions and we would like address our deeply gratitude.

Sincerely,

Correspondence to:

Yang Xiang, MD. Professor, Peking Union Medical College Hospital, Chinese Academy of Medical Sciences, Peking Union Medical College, No.1 Shuai fu yuan Wang fu jing, Dong cheng District, Beijing, 100730, People’s Republic of China.

Email: [email protected].

Tel: 0086-01065296068

Fax: 0086-01065296218